# Large-Scale Detection and Categorization of Oil Spills from SAR Images with Deep Learning

**Filippo Maria Bianchi** [1,2,*] , **Martine M. Espeseth** [2] **and Njål Borch** [1]

1    NORCE the Norwegian Research Centre AS, 5008 Bergen, Norway; njbo@norceresearch.no
2    Department of Physics and Technology, UiT the Arctic University of Norway, 9019 Tromsø, Norway; martine.espeseth@uit.no
*    Correspondence: fibi@norceresearch.no

**Abstract:** We propose a deep-learning framework to detect and categorize oil spills in synthetic aperture radar (SAR) images at a large scale. Through a carefully designed neural network model for image segmentation trained on an extensive dataset, we obtain state-of-the-art performance in oil spill detection, achieving results that are comparable to results produced by human operators. We also introduce a classification task, which is novel in the context of oil spill detection in SAR. Specifically, after being detected, each oil spill is also classified according to different categories of its shape and texture characteristics. The classification results provide valuable insights for improving the design of services for oil spill monitoring by world-leading providers. Finally, we present our operational pipeline and a visualization tool for large-scale data, which allows detection and analysis of the historical occurrence of oil spills worldwide.

**Keywords:** oil spills; deep learning; SAR; object detection; image segmentation

## 1. Introduction

Spaceborne Synthetic Aperture Radar (SAR) instruments have been used for monitoring and early detection of oil spills for several decades and are a well-established tool for many operational monitoring services (see e.g., [1,2]). This information is paramount for planning oil spill preparedness strategies where location, extent, and early warning are relevant. Oil detections can concern legal, illegal, and accidental discharge from offshore installations, ships, and pipelines. For example, in the North sea, oil slicks resulted from discharge of produced water (produced water contains a small amount of mineral oil that consists of various constituents depending on the age and the type of oil well [3,4]) are frequently detected around oil platforms [3–5]. The large amount of data archived over the years of detected and (sometimes) verified oil spills, is a direct result of the long-term use of SAR. This has created a huge potential for developing new technologies and methods, which can take advantage of this work-effort and archived data. Remarkably relevant is the free database of satellite images obtained from the Sentinel-1 sensors, which is acquiring images since April 2014 (Sentinel-1A) and April 2016 (Sentinel-1B) [6]. SAR sensors allow monitoring the surface independently of the weather and sun conditions. This is especially important in the North Sea and the Barents Sea, due to frequent cloud coverage and darkness for long periods of the year.

In SAR images, the oil slicks appear as dark patches due to the low backscatter response compared to the surrounding clean sea areas. The low backscatter is a result of the oil damping of short-gravity and capillary ocean surface waves. The dark signature of oil slicks in SAR is also common for many other ocean features, such as low-wind areas and natural oil slicks, which are also known as *look-alikes*. An extensive effort has been made to design methodologies to distinguish oil slicks from natural biogenic slicks and/or low-wind areas (see e.g., [1,7–11]).

The backscatter of oil slicks depends on several factors like wind, sensor properties, and oil characteristics. For example, the wind is the main factor generating the ocean surface roughness and, therefore, wind significantly affect the oil-sea contrast. The oil-sea contrast also depends on the incidence angle of the satellite, as it affects the backscatter response. In particular, both high and low incidence angles yield low oil-sea contrast [12,13]. All these factors make it challenging to automatically detect, segment, and classify oil slicks.

There are several works on oil slick segmentation and classification using SAR. Segmenting oil slicks or dark objects in SAR imagery has been performed for several decades, and most of the traditional oil slick classification algorithms (see, e.g., [1,9,11,14]) consist of three stages; (1) detection of low-backscattering targets; (2) feature selection; (3) statistical classification methods. A thorough review of the traditional and the early work regarding oil slick segmentation and classification methods was presented in 2005 [2]. Early work by [14] on distinguishing oil spills from look-alikes used a multilayer perceptron together with a visual inspection for areas of interest followed by features extraction. Topouzelis et al. [11] used two neural networks for segmenting dark objects and then separating potential oil slicks from look-alikes, hence, avoiding visual inspection of the selected area. This framework has been adopted in several later studies [9,15].

The application in remote sensing of deep-learning models for computer vision outperformed the previous signal processing techniques and sets the new state-of-the-art in several tasks [16]. In the last few years, several works proposed to use convolutional neural networks (CNNs) to detect oil spills [10,17,18]. Compared to traditional pattern recognition approaches, CNNs can be trained end-to-end, meaning that they learn from examples how to map input data into the desired output [19]. First, this greatly simplifies the task of the practitioner that is not required to design rules and specify critical hyperparameters (e.g., thresholds) to solve the inference task and that generalize well to unseen data. Second, the practitioner is relieved from hand-crafting features that are, instead, automatically learned by a CNN by optimizing the training objective. Indeed, human-engineered features come with biases and are hampered by the limitations of humans in discovering complicated patterns and relationships in the data [20]. On the other hand, deep-learning models are exceptionally data-hungry, especially if the architectures are large and have many trainable parameters. To learn features and classification rules that are general enough and do not overfit the training data, it is necessary to expose the model to a large amount of input-output pairs, which are SAR images and segmentation masks in the case of oil spills detection. Unfortunately, while unlabeled data are cheap and available in large quantity, labels are usually scarce and costly to obtain [21]. This is the reason there are no examples of deep-learning models trained on a large-scale dataset for oil spill detection.

The main objective of this paper is to develop a robust and automated framework to detect and classify oil slicks that will benefit operational monitoring services and oil spill preparedness authorities. Our contributions are summarized as follows.

- (*Detection*): we develop a deep-learning architecture that detects oil spills in SAR scenes with high accuracy. When trained on a large-scale dataset, our model achieves extremely high performance.
- (*Categorization*): each oil spill detected by our deep-learning model is further processed by a second neural network, which infers information about shape, contrast, and texture.
- (*Visualization*): we present our production pipeline to perform inference at a large scale and visualize the obtained results. Our visualization tool allows analyzing the presence of oil spills worldwide during given historical periods.

Closely related to our first contribution (detection), is the work presented in [10] that compares the performance of six existing CNN models for semantic segmentation in performing oil spill detection. Among the tested models, DeepLabv3+ [22] achieves the best segmentation performance. To train and evaluate the models, the authors introduce a new segmentation dataset based on the pollution events provided by the European Maritime Safety Agency (EMSA) through the CleanSeaNet service. Compared to ours, the dataset is smaller as it consists of 1112 images, each one covering an area of approximately $(12.5 \times 6.5)$ km$^2$. Additionally, the images are associated with segmentation masks

with 5 classes (sea, land, oil, ships, and look-alikes), while we use binary masks (oil and non-oil). In particular, in our dataset sea, land, ships, and look-alikes are all assigned to the same label (non-oil). Hence, this study focuses only on detecting mineral oil slicks, rather than natural slicks (look-alikes). Finally, differently from [10], we do not adopt off-the-shelf architectures but rather propose a CNN model, a training, and evaluation procedure that are optimized for the oil spill segmentation task at hand.

The remainder of this paper is organized as follows. In Section 2, we describe the original data and the preparation of the dataset used for training the proposed deep-learning framework. In Section 3, we present the CNN architecture used to perform semantic segmentation of oil spills. Section 4 describes a second CNN model that categorizes the oil spills after they are detected by the segmentation neural network. In Section 5 we report the performance of the proposed framework and discuss the results obtained in comparison to the golden standard of manual labeling. Section 6 describes the software we developed for large-scale visualization and analysis, based on the proposed deep-learning framework. Finally, in Section 7 we draw our conclusions.

## 2. Dataset Description

The dataset has been produced by Kongsberg Satellite Service (KSAT) (www.ksat.no) and consists of Synthetic Aperture Radar (SAR) scenes from Sentinel-1, associated with a binary mask generated by trained human operators at KSAT, which indicates the location of the oil spills. KSAT has a long and consolidated experience in oil detection by SAR, offering worldwide near-real-time services with extensive coverage and fine temporal resolution based on different SAR sensors. The masks associate each pixel in the SAR image with a label that is 0 for the class "non-oil spill" and 1 for the class "oil spill". The "oil spill" class is assumed to only indicate mineral oil slicks from e.g., legal and illegal release. On the other hand, the "non-oil spill" includes clean sea, land, ships, and look-alikes (low-wind areas, biogenic slicks etc.).

The whole dataset consists of 713 products of the Sentinel-1 sensor and each product covers an area up to approximately 150,000 km$^2$. The SAR products are relative to regions in northern Europe, in particular the Norwegian Sea, the Barents Sea, the North Sea, and the Baltic Sea. The products are collected over a period of 4 years between 2014 and 2018, and they contain 2093 *oil spill events*. An oil spill event refers to a collection of one or several individual oil slicks that are located nearby and originate from the same source, according to trained human operators at KSAT. The total number of individual oil slicks is 227,964.

The SAR scenes are acquired with the dual-polarimetric SAR mode (IW mode of Sentinel-1) using the vertical transmit and vertical receive (VV) and vertical transmit and horizontal receive (VH) polarization channels with the medium resolution mode (see [6] for additional information about Sentinel-1). The VV polarization is preferred over VH for oil spill detection, due to less impact of system noise in VV compared to the VH channel. For this reason, the VV channel is the one considered in this study.

All the SAR products used for training the deep-learning models are smoothed for noise removal and are at 40 m resolution, i.e., each pixel covers an area of $40 \times 40$ m. After the model is trained, detection can also be performed on high-definition (10 m resolution) SAR products that are publicly available from the Copernicus Open Access Hub (https://scihub.copernicus.eu/). When processing high-definition SAR images, we first apply a boxcar filter with size 11 and then we downsampled the images to 1/4 of their original size to recover the 40m resolution.

The values in the VV channel are provided in 16bit unsigned integer format, meaning that the backscatter assumes values in the interval $[0, 2^{16}]$. Radiometric calibration to sigma-nought is not performed since is a linear transformation and we expect the deep-learning model to learn to apply such a transformation if needed. Very few outlier pixels in the dataset have high backscatter values up to $2^{16}$ (typically backscattering from ships/platforms or pixels located at high incidence angle), while the majority of pixels have backscatters concentrated around a much smaller interval.

Figure 1 shows a normalized distribution of the values of the pixels marked as "oil" and "non-oil" computed over all the SAR products in our dataset. Since pixels with high backscatter are very few, for visualization purposes the density plots show a limited range $[0, 500]$ of backscatter values, rather than the whole interval $[0, 2^{16}]$.

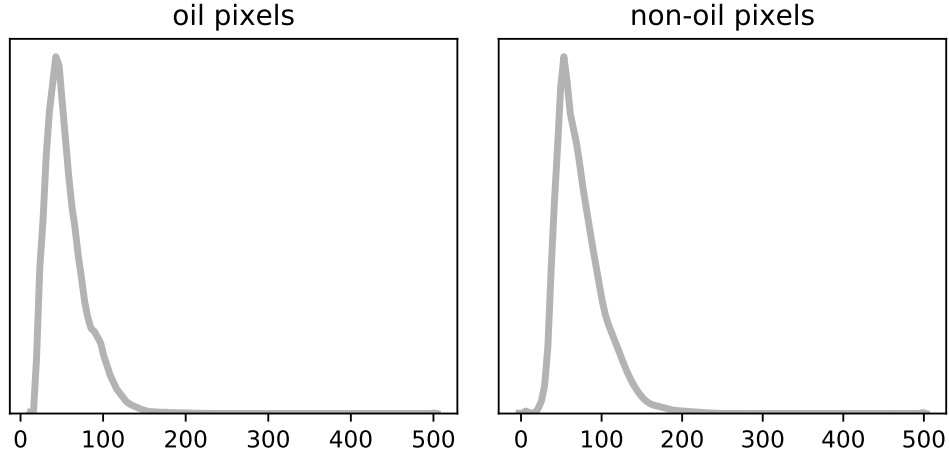

**Figure 1.** Distribution of the backscatter values in the VV images, across pixels belonging to class "oil" and "non-oil". The plots show only the range $[0, 500]$, rather than $[0, 2^{16}]$.

As we can see from the plots, the backscatter distribution is shifted toward small values in the interval. As expected, the backscatter distribution of oil-covered pixels is shifted towards values that are lower than non-oil pixels. Based on this analysis, we clip the maximum value in all VV images in the dataset to 150, which corresponds to approximately the 98% percentile. Notably, all the oil spill pixels have a backscatter value lower than 150.

Besides the binary masks indicating the position and shape of oil spills, each oil spill event in the dataset is categorized according to 12 different fields, which indicate the type of shape and texture of the oil spill. Table 1 reports the name of the 12 categories, the set of possible values assumed by each category, and how such values are distributed across the dataset elements. It is possible to notice immediately that for several categories, such as the texture attributes, the distribution of the values is very skewed. This likely indicates that discriminating across certain categories is challenging for the human operators that are labeling the SAR scenes. In particular, texture categories are difficult to detect given the inherent noise in the VV channel. An objective of this study is to assess if such categories can be predicted using machine learning; indirectly this indicates how much information about texture, shape, and contrast can be extrapolated from the SAR image.

**Table 1.** The 12 categories used to classify each oil spill. The possible values assumed by each category are reported in the second column. The distribution of the values for each category assumed by the elements in the dataset is reported in the third column, by using both numerical percentages and a histogram; e.g., 61.8% of the data samples do not have a linear shape and 38.2% have a linear shape.

| Category | Values | Values Distribution | |
|---|---|---|---|
| Patch shape | {False, True} | {48.2%, 51.8%} | |
| Linear shape | {False, True} | {61.8%, 38.2%} | |
| Angular shape | {False, True} | {90.4%, 9.6%} | |
| Weathered texture | {False, True} | {71.3%, 28.7%} | |
| Tailed shape | {False, True} | {83.2%, 16.8%} | |
| Droplets texture | {False, True} | {98.3%, 1.7%} | |
| Winding texture | {False, True} | {92.5%, 7.5%} | |
| Feathered texture | {False, True} | {97.7%, 2.3%} | |
| Shape outline | {Fragmented, Continuous} | {78.4%, 21.6%} | |
| Texture | {Rough, Smooth, Strong, Variable} | {29.2%, 14.3%, 5.1%, 51.2%} | |
| Contrast | {Strong, Weak, Variable} | {23%, 52.8%, 24.1%} | |
| Edge | {Sharp, Diffuse, Variable} | {33.7%, 13.3%, 52.9%} | |

## 2.1. Division in Patches

To convert the dataset in a format suitable for training a CNN architecture, we extracted a set of patches of size $160 \times 160$ pixels from the SAR products, each one covering an area of 41 km$^2$. We note that it would be possible considering larger patches to capture more contextual information in the images. However, this would be traded with a longer training time, which in our case is already in the order of months (see Section 5.1.4), as the deep-learning models could be fed with smaller batches of samples during training. Also, smaller patches yield more training samples, which allows limiting overfit by injecting more stochasticity in the training process. Finally, related works often use SAR data with higher resolution (e.g., 10m pixel-resolution [10]), while we consider 40m resolution that allows us to cover larger areas with a smaller patch.

An entire oil spill event can be very large and a patch, in general, does not cover it completely. We recall that an oil spill event can consist of several individual neighboring oil slicks originating from the same source. By referring to the example in Figure 2, 8 patches are necessary to cover the oil event depicted in the figure. Since the labels in Table 1 are associated with a whole oil event, which can be composed of multiple oil slicks, there are two options to associate the labels to the patches. The first is to assign the same label to all the patches covering the same oil event. The second is to take the most central patch covering the oil event (depicted as the green box in Figure 2) and the label of the whole oil event is assigned only to such a central patch. We opted for this second option and ended up with a total of 2093 "centered" patches associates with a label describing the values of the 12 attributes in Table 1. We refer to this dataset as $\mathcal{D}_1$.

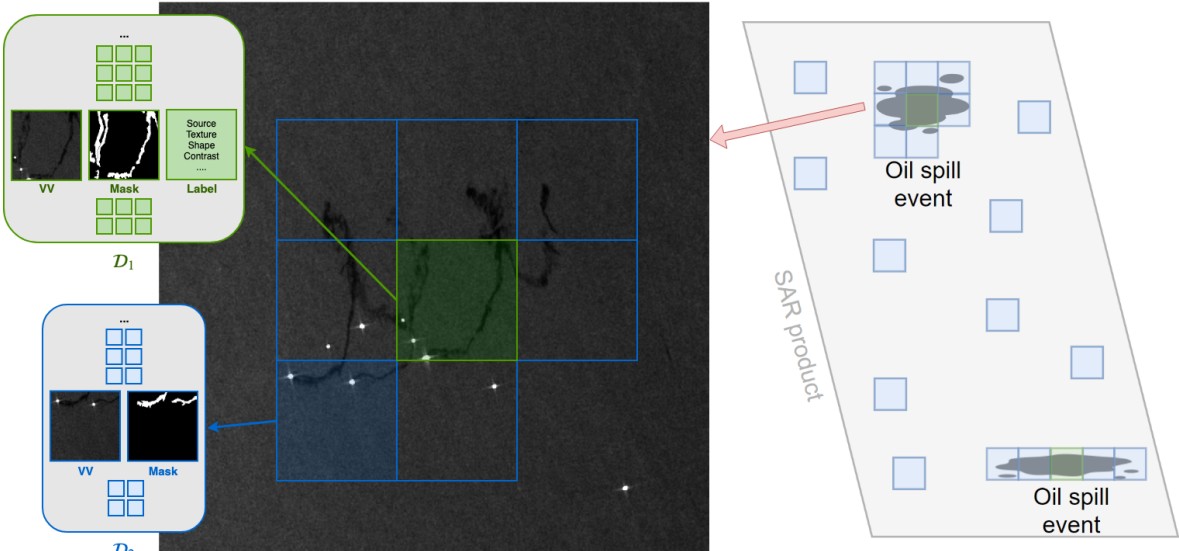

**Figure 2.** Illustration of patches extraction from the SAR products. The patches centered on the oil spill events (depicted in green) form the dataset $\mathcal{D}_1$ and are associate with a label that indicates the values assumed by the 12 categories. The second dataset, $\mathcal{D}_2$, contains (i) all the patches of $\mathcal{D}_1$, (ii) all the patches with at least 1 oil spill pixel, (iii) an equal amount of patches without oil that is randomly sampled from other locations in the SAR product. Along with the VV channel, the segmentation masks are included in both datasets.

We note that $\mathcal{D}_1$ includes only a fraction of the available segmentation masks: by referring to the oil event depicted in Figure 2, all the data outside the green central patch is not contained in $\mathcal{D}_1$. To exploit also the remaining information (i.e., the segmentation masks associated with the oil spill pixels outside the green box in Figure 2), we built a second dataset $\mathcal{D}_2$, which includes all the patches of $\mathcal{D}_1$ plus all the patches containing at least one pixel belonging to the oil class. The additional patches assigned to $\mathcal{D}_2$ are depicted as blue boxes in Figure 2. We stress that all the patches in $\mathcal{D}_1$ (green boxes) are also included in $\mathcal{D}_2$. During the training phase, we want to expose the segmentation model also to patches where no oil pixels are present. Therefore, we also included in $\mathcal{D}_2$ the patches without any oil spill pixels, which are randomly sampled from the SAR products. The total number of patches in $\mathcal{D}_2$ is $187,321$ and approximately half of the patches do not contain any oil pixel. The total amounts of pixels belonging to class 0 and 1 are $59,689,609$ and $1,243,242$, respectively. Therefore, the pixels of class "oil" are 2.04% of the total.

*2.2. Division on in Training, Validation, and Test Set.*

The dataset $\mathcal{D}_1$ is used to train the deep-learning model that performs categorization (see Section 4) and to perform hyperparameter selections (see Section 5). We split $\mathcal{D}_1$, in training, validation, and test set with sizes 1843, 150, and 100 respectively.

The dataset $\mathcal{D}_2$ is used to train the model that performs oil spills detection and is split in a training and validation set of sizes 149,856 and 37,465, respectively. We ensured that all the patches in the validation and test set of $\mathcal{D}_1$ are excluded from the training set of $\mathcal{D}_2$.

From the original dataset consisting of 713 SAR products, 3 entire products are kept aside, i.e., they are not used to extract the patches that populate $\mathcal{D}_1$ or $\mathcal{D}_2$. These 3 products form a separate dataset, $\mathcal{D}_{\text{test}}$, which is used exclusively to test the performance of the segmentation task. Table 2 summarizes the content of the datasets $\mathcal{D}_1$, $\mathcal{D}_2$, and $\mathcal{D}_{\text{test}}$. Table 3 provides additional details on $\mathcal{D}_{\text{test}}$, which will be useful for discussing the results in Section 5.

**Table 2.** Summary of the datasets details.

| Original Data | $\mathcal{D}_1$ | $\mathcal{D}_2$ | $\mathcal{D}_{\text{test}}$ |
|---|---|---|---|
| • 713 SAR prod. <br> • 4 years period <br> • 2,093 oil events <br> • 227,964 oil spills | • 1843 tr. patches <br> • 150 val. patches <br> • 100 test patches | • 149,856 tr. patches <br> • 37,465 val. patches | • 3 SAR products <br> • Details in Table 3 |

**Table 3.** Further details on $\mathcal{D}_{\text{test}}$, i.e., the three SAR products used as test set. The image size is reported as the number of pixels.

| ID | Image Size | # Oil Spills | # Oil Pixels | # Non-Oil Pixels |
|---|---|---|---|---|
| T1 | $9836 \times 14{,}894$ | 2 | 552 (0.00067%) | 81,877,550 |
| T2 | $9470 \times 21{,}738$ | 11 | 19,336 (0.017%) | 112,071,385 |
| T3 | $10{,}602 \times 21{,}471$ | 36 | 22,793 (0.018%) | 127,436,689 |

## 3. Oil Spill Detection

Oil spill detection is conveniently framed as a semantic segmentation task, which consists of performing pixel-level classification [23]. In the following, we first describe the neural network model used to perform segmentation. Then, we present the procedures adopted for training the model and to perform inference on new, unseen data.

### 3.1. The Deep-Learning Architecture for Semantic Segmentation

The model used to perform segmentation is a fully convolutional network, referred in the remainder of the paper as *OFCN* (Oil Fully ConvNet). The OFCN is a network with no dense layers that, therefore, can process inputs of variable size. This allows for training on small images and processing larger ones at inference time. The OFCN model is based on the U-net [24], a popular deep-learning architecture for image segmentation that is also used in several remote sensing applications [25–27].

The OFCN consists of an *encoder* and a *decoder* part, respectively depicted in green and purple in Figure 3. The encoder gradually extracts feature maps that detect the patterns of interest in the image. By reducing the spatial dimensions and increasing the number of filters, the deeper layers in the encoder capture features of increasing complexity and larger spatial extent in the input image. The decoder gradually transforms the high-level features and, in the end, maps them into the output. The output is a binary segmentation mask, which has the same height/width of the input image and associates to each pixel a class value: 1 if it belongs to the oil class, 0 otherwise. The skip connections link the feature maps from the encoding to the decoding layers, such that some information can bypass the bottleneck located at the bottom of the architecture. In this way, our architecture still learns to generalize from the high-level latent representation but also recovers spatial information from the intermediate representations through a pixel-wise semantic alignment.

Each block in the encoder consists of 9 layers, as depicted in the green box of Figure 3 (bottom-left). Conv($n$) indicates a convolutional layer with $n$ filters (e.g., $n = 32$ in the 1st block, 64, 128, 256, and 512 in the 2nd, 3rd, 4th and 5th blocks, respectively), filter size $3 \times 3$, stride 1, and padding modality *same* [28]. Each Conv layer is followed by a Batch Normalization (BN) layer [29] and a ReLU activation function. At the end of each block, there is a max-pooling with stride 2, a Squeeze-and-Excitation (SE) [30], and a Dropout layer [31]. Each unit in the deeper layer of the encoder has a receptive field of 140, meaning that each feature depends on a neighborhood with a radius of 140 pixels in the input image. A visualization of the growth of the receptive field in the encoder layers is reported in Appendix A.

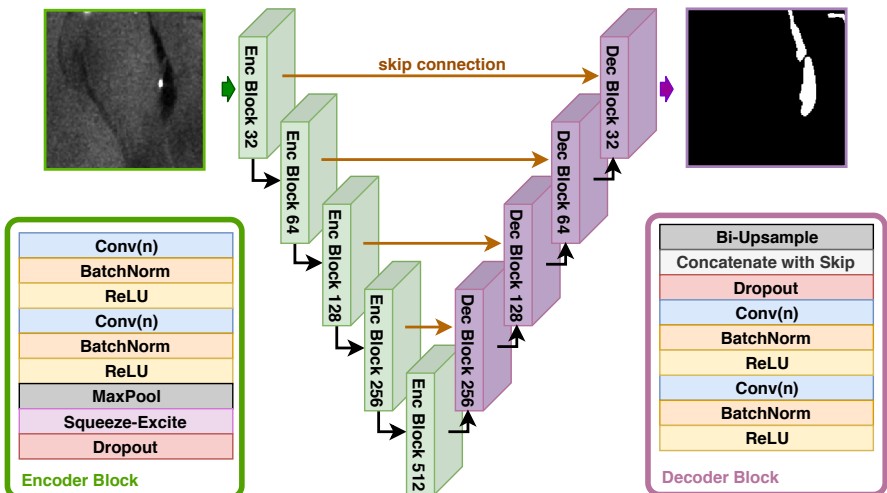

**Figure 3.** Schematic depiction of the OFCN architecture used for segmentation. Conv(*n*) stands for a convolutional layer with *n* neurons. For example, *n* = 32 in the first Encoder Block, 64 in the second, and so on.

Compared to the encoder, the decoder has a somehow mirrored structure and the details of its blocks are shown in Figure 3 (bottom-right). The main differences are i) the Bilinear upsampling layers, which upscale the features map of the previous layer, and ii) the concatenation with the skip connections that inject in the decoder the output of the encoder blocks. Finally, the last decoder block replaces the second ReLU activation with a sigmoid that produces output in the interval $[0, 1]$. This is a common choice in binary classification tasks, such as the generation of the binary oil spill mask.

Let *n* be the number of convolutional filters in the first layer, the number of filters in the rest of the OFCN architecture is univocally determined and is $n$-$(n \times 2)$-$(n \times 4)$-$(n \times 8)$-$(n \times 16)$-$(n \times 8)$-$(n \times 4)$-$(n \times 2)$-$n$. For conciseness, we will use the notation OFCN(*n*) in the rest of the paper. For example, the architecture in Figure 3 is OFCN(32).

A detailed description of the Bilinear upsampling, Batch Normalization, Squeeze-and-Excitation, and Dropout modules is deferred to the Appendix.

### 3.2. Training and Inference

The OFCN is trained to predict the binary segmentation mask, indicating the location of the oil spills. The network weights are optimized by iteratively minimizing a loss function evaluated on mini-batches of input-output pairs. The choice of the loss function, the training and the test procedures are reported in the following.

#### 3.2.1. Loss Functions for Unbalanced Dataset

Oil spills are small objects and, as discussed in Section 2, they represent only a tiny fraction (≈2%) of the entire dataset. Due to the strong imbalance between the pixels of class 0 ("non-oil spill") and class 1 ("oil spill"), a naive classifier can achieve an accuracy of ≈98% simply by assigning all the pixels to class 0.

To handle the class imbalance, rather than training the OFCN with the standard binary cross-entropy loss, a binary cross-entropy with class balancing is used. Specifically, the standard cross-entropy is re-weighted to assign a larger penalty when a pixel of the under-represented class is wrongly classified. This can be done by weighting the loss associated with each pixel of the oil class with a value larger than for the non-oil class. We also tested three additional loss functions that account for class imbalance, but we obtained unsatisfactory results. The details are reported in Appendix C.

### 3.2.2. Data Augmentation

To prevent the model from overfitting the training data and to enhance its generalization capability on unseen examples, we augment the dataset during training. Randomized data augmentation can improve the generalization performance in several computer vision tasks, including applications on remote sensing [32]. In particular, we apply the following random transformations on the fly: horizontal and vertical flips, horizontal and vertical shifts, rotations, zooming and shearing to the training images. To ensure consistency between input and the target segmentation masks used for training, the same transformations of the input are also applied to the oil masks.

### 3.2.3. Two-Stage Training

We first trained the OFCN on a low-resolution version of the dataset, obtained by downsizing the patches size by half ($80 \times 80$ pixels). After having completed the training on the low-resolution dataset, we resumed the training on the patches of original size *without resetting the network weights*. This is possible because, as discussed in Section 3.1, the OFCN can consume images of variable size, since its weights are independent of the input shape.

The intuition behind training first on downsized images is to let the model learn first the coarser structure in the inputs and then refine the parameters' tuning as the incoming images expand and become more detailed. Indeed, the first training phase allows the quick bringing of the OFCN on a good initial configuration, which serves as a starting point to find a better solution when training on the higher-resolution images. Similar in spirit to our idea is the approach adopted by the Progressive Growing of GANs [33].

### 3.2.4. Test Time Augmentation.

When computing the prediction of a SAR scene at inference time, we *slide* the OFCN on the large image, computing predictions for one window at a time. Again, we stress that the window size at test time can be larger than $160 \times 160$ pixels. However, this approach usually generates checkerboard artefacts and border effects close to the window edges. To obtain smoother and more accurate predictions, overlapping sliding windows with a stride equal to half the window size are processed by the OFCN. Furthermore, 8 predictions from all the possible 90° rotations and flips of each window are generated. To obtain the final result, we used a $2^{nd}$ order spline interpolation to join all the computed predictions.

## 4. Oil Spill Categorization

With *oil spill categorization* we refer to the task of inferring the 12 categories (described in Table 1), which indicate the texture, shape, and contrast of the oil spill. Such categories are useful to end-users and analysts in deriving information about the source, stage of weathering and internal variations within the oil slicks. To predict each of the 12 categories, we train a separate instance of the architecture depicted in Figure 4.

The categorization model takes as input a SAR patch and the associated mask predicted by the OFCN model described in Section 3. The (predicted) segmentation mask encourages the categorization network to focus on the areas of the patch with oil spills and allows more easy extraction of shape information.

Figure 4 shows the whole pipeline: input SAR image → detection → categorization. The pipeline is not trained end-to-end. In fact, we first train the OFCN and, afterwards, the categorization network. The first, obvious reason for not training the whole pipeline end-to-end is that the category labels are available only for $\mathcal{D}_1$ and not $\mathcal{D}_2$, the large dataset that is used to train the OFCN. Moreover, compared to the segmentation masks, the 12 category labels are noisier as they are more subjective to human interpretation. Therefore, to achieve the best possible segmentation performance, the OFCN is trained independently without being conditioned by the categorization loss.

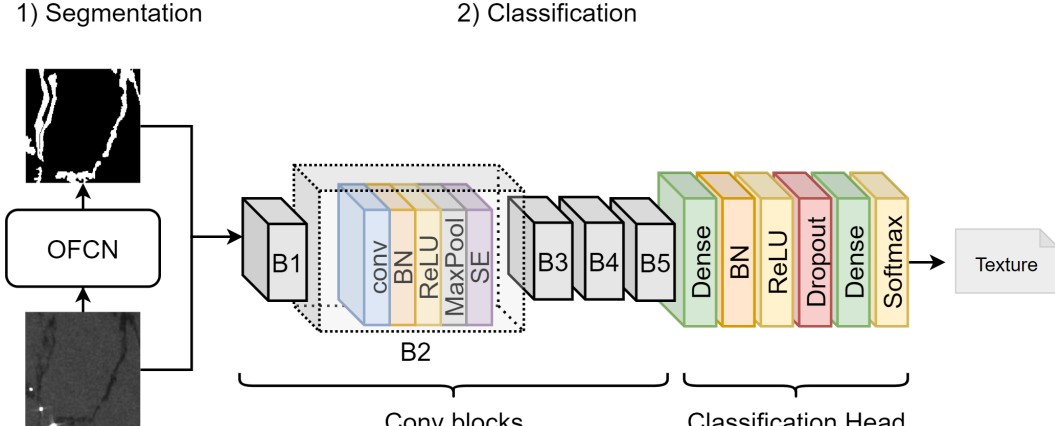

**Figure 4.** First the trained OFCN generates the segmentation masks from the SAR images. Then, both SAR images and predicted mask are fed in the classification network. A different architecture is trained to classify each one of the categories (e.g., the depicted one classifies the "texture" category).

The categorization network in Figure 4 consists of 5 convolutional blocks responsible for feature extraction. Each block is structured as [conv($n$)-BN-ReLU-Maxpool-SE], where the number of convolutional filters $n$ is 32, 64, 128, 256, and 512 in the blocks B1-B5, respectively. The classification head has the following architecture: Dense(256)-BN-ReLU-Dropout-Dense(#cat)-Softmax. The layers "Dense" are two fully connected layers whose numbers of units are 256 and the number of values assumed by each category (#cat), respectively. The network is trained by minimizing a categorical cross-entropy loss and by using the same image augmentation procedure used for training the OFCN architecture.

We note that the task of distinguishing mineral oil slicks from look-alikes (often referred to as oil slick classification in the literature) is not performed in our study as no such information was contained in our dataset, although look-alikes are present in the "non-oil" class. The proposed categorizing scheme presented here extracts information about the oil slicks involving contrast, shape, and texture, that could be useful to distinguish look-alikes (low-wind areas or biogenic slicks) from mineral oil slicks as different contrast, shape and texture might be expected. Look-alikes stem from natural factors (wind, seep, algae, etc.), while human-made oil slicks originate from either legal, illegal, or accidental releases from platforms, ships, or pipelines. The origin of the different slicks in the sea is challenging to infer only by using SAR data, and complementary information about the site (sea maps, wind history, algae events) is necessary.

## 5. Results and Discussion

In this section, we describe the experimental setting and report the results obtained for the detection and the categorization task, respectively.

### 5.1. Oil Spill Detection: Experimental Setting, Analysis, and Results

5.1.1. Evaluation Metrics

While the parameters of OFCN are optimized by minimizing the loss described in Section 3, we require more interpretable metrics to quantify the results obtained on the test set, and also to monitor the model performance on the validation set during training. We consider two metrics different than accuracy, since the oil class is highly under-represented in the segmentation task and, similarly, some categories are imbalanced in the categorization task.

The first metric is the F1 score, which is computed at the pixel level and is defined as

$$F1 = 2 \frac{\text{precision} \cdot \text{recall}}{\text{precision} + \text{recall}},$$

where *precision* is defined as $\frac{TP}{TP+FP}$ and *recall* is $\frac{TP}{TP+FN}$ (TP = True Positives, FP = False Negatives, FN = False Negatives). To compute the F1 score in the segmentation task, the output $o$ of the sigmoid in the last layer of the OFCN is rounded as follows: $o = 1$ if $o \geq \tau$, $o = 0$ if $o < \tau$. When not specified otherwise, in the rest of the paper we assume that $\tau = 0.5$ is used in the rounding procedure.

In each experiment presented in the following, the F1 score on the validation set is used to evaluate during training the performance of the current model on unseen data. Specifically, whenever the model improves its F1 score on the validation set, the current instance of the weights is saved as the best model.

We also consider a second metric that indicates if the OFCN managed to correctly locate the oil spill, without accounting for small differences in the shape contours between human-made and predicted segmentation masks. For this purpose, we consider the bounding boxes that contain oil spills in both the human-made and predicted mask. The bounding boxes are obtained with standard image processing libraries and are computed "offline" on the segmentation masks predicted by trained OFCN and after thresholding. To quantify how much the bounding boxes in the ground truth and the prediction overlap, we computed the intersection over union (IoU):

$$IoU = \frac{\text{Area of bounding boxes intersection}}{\text{Area of bounding boxes union}}.$$

Contrarily to the F1 score that is evaluated during training to save the best model, the IoU is only computed once the training is over to evaluate the final performance.

### 5.1.2. Hyperparameters Search

To find the optimal configuration of the hyperparameters in the OFCN, we performed cross-validation by randomly sampling configurations from the hyperparameters space and selecting those that yield the highest performance on the validation set. A total of 500 configurations are randomly sampled from the following hyperparameters space: BN {True, False}, SE {True, False}, Loss {weighted binary cross-entropy, Jaccard, Focal, Lovász-softmax}, $L_2$ penalty {0, $1 \times 10^{-6}$, $1 \times 10^{-5}$, $1 \times 10^{-4}$, $1 \times 10^{-3}$, $1 \times 10^{-2}$}, Dropout {0, 0.1, 0.25, 0.5}, learning rate of Adam [34] optimizer {$1 \times 10^{-4}$, $5 \times 10^{-4}$, $1 \times 10^{-3}$, $5 \times 10^{-3}$}, oil class weight (only binary cross-entropy) {1, 2, 3, 5}. We also tried weights higher than 5 for the oil class, but obtained a very large number of false positives. To make the hyperparameters search tractable, we used a smaller architecture, OFCN(16), and we trained it using the training and the validation set of $\mathcal{D}_1$ (which are much smaller than $\mathcal{D}_2$) for 100 epochs only. We used mini-batches of size 32 and image augmentation is used with the following parameters: max rotation $90°$, max-width shift 0.1 of total width, max height shift 0.1 of total height, max shearing 0.3, max zoom 0.2, probability of horizontal and vertical flips 0.5, pad mode "mirror". Table 4 reports the F1 score obtained by the 3 best configurations.

**Table 4.** Hyperparameters selection results. We report the 3 best configurations (C1, C2, C3) found with cross-validation on $\mathcal{D}_1$. *Acronyms:* BN (Batch Normalization), SE (Squeeze-and-Excitation), $L_2$ reg. (strength of the $L_2$ regularization on the network parameters), LR (Learning Rate), CW (weight of the oil class).

| ID | BN | SE | $L_2$ reg. | Dropout | LR | CW | Var F1 ($\mathcal{D}_1$) |
|----|------|-------|-----------------|---------|-------------------|----|--------------------------|
| **C1** | True | True | 0.0 | 0.1 | $1 \times 10^{-3}$ | 2 | **0.731** |
| **C2** | False | True | $1 \times 10^{-6}$ | 0.1 | $5 \times 10^{-4}$ | 3 | 0.723 |
| **C3** | True | False | 0.0 | 0.0 | $1 \times 10^{-3}$ | 2 | 0.708 |

### 5.1.3. Comparison with Baselines

We compare the performance of the proposed OFCN with the vanilla U-net architecture [24] and with DeeplabV3+ [22], which is considered, at the time of writing, the state-of-the-art for

image segmentation. Notably, DeeplabV3+ is the segmentation architecture that achieved the best performance in related work on oil spill segmentation in Ref. [10].

To perform the comparison, we used the Keras implementations of U-net and DeeplabV3+ available at two popular public repositories (U-net: https://github.com/zhixuhao/unet, DeeplabV3+: https://github.com/bonlime/keras-deeplab-v3-plus). For this experiment, we used the larger OFCN(32) architecture with configuration C1 described in Table 4. The DeeplabV3+ is configured with the Xception backbone [35]. All the settings are the same as in the previous experiment, with the exception that the models are trained for 400 epochs and the batch size is 16 (The DeeplabV3+ is a network with very high capacity and, during training, a batch of 32 patches does not fit in the GPU).

**Table 5.** Comparison with baselines. Reported is the number of trainable parameters, training time for 400 epochs, training accuracy, training loss, and F1 score on the validation. Best results are in bold. Models are trained on an Nvidia RTX 2080.

| Model | # Params. | Tr time (hours) | Tr Acc. | Tr Loss | Val F1 ($\mathcal{D}_1$) |
|---|---|---|---|---|---|
| U-net | 7,760,069 | **10.1** | 0.984 | 0.058 | 0.741 |
| DeepLabV3+ | 41,049,697 | 15.2 | 0.987 | 0.039 | 0.765 |
| OFCN | 7,873,729 | 10.9 | **0.988** | **0.038** | **0.775** |

Table 5 reports the number of trainable parameters in each architecture, the time (in hours) necessary to complete 400 epochs of training, the final training accuracy and training loss, and the best F1 score obtained on the validation. First, we notice that DeeplabV3+ has much more trainable parameters compared to OFCN and U-net, which makes its training almost 50% slower than for the other two architecture. On the other hand, the training times of OFCN and U-net are comparable. DeeplabV3+ outperformed U-net but, despite its larger capacity, did not achieve a better performance than the proposed OFCN architecture.

Importantly, we report that in some runs the U-net did not manage to learn anything: the loss was not decreasing and the predicted output was a mask of all zeros for each image in the training set. This suggests a strong sensitivity to the weight initialization and a jagged loss landscape, where the optimizer might get stuck in local minima. Finally, we also experimented with DeeplabV3+ configured with the Mobilenet [36] backbone but we obtained unsatisfactory performance.

The training graphs showing the evolution of the loss on the training set and the F1 score on the validation set are in Appendix B and they show that none of the models overfits the training data.

### 5.1.4. Training on the Large-Scale Dataset

First, we trained the three OFCN(32) models configured with the best hyperparameters settings (C1, C2, C3 reported in Table 4) for 400 epochs. Training each model on the large-scale dataset $\mathcal{D}_2$ takes up to one week on an Nvidia RTX 2080. The results reported in Table 6 confirms that the OFCN configured with C1 obtains the best performance: highest accuracy and lowest loss on the training set, highest F1 score on the validation set.

**Table 6.** Validation performance obtained on $\mathcal{D}_2$ using the three best configurations C1, C2, and C3, the two-step training (2ST) strategy with configuration C1, and a long training of 3000 epochs (Long). We also report the training time, the accuracy, and loss achieved on the training set.

| ID | Epochs | Time (days) | Tr Acc. | Tr loss | Val F1 ($\mathcal{D}_2$) |
|---|---|---|---|---|---|
| C1 | 400 | 6.8 | 0.995 | 0.016 | 0.857 |
| C2 | 400 | 6.9 | 0.990 | 0.047 | 0.750 |
| C3 | 400 | 6.2 | 0.993 | 0.018 | 0.802 |
| C1-2ST | 400 + 400 | 9.4 | 0.996 | 0.014 | 0.861 |
| C1-2ST-Long | 500 + 3000 | 54.3 | 0.997 | 0.009 | **0.892** |

Then, we applied the 2-stage training strategy (see Section 3.2), using the configuration C1 (C1-2ST in Table 6). The procedure takes approximately 50% extra time since in the first stage the images are downsampled with a factor of 2 but it yields some performance improvement.

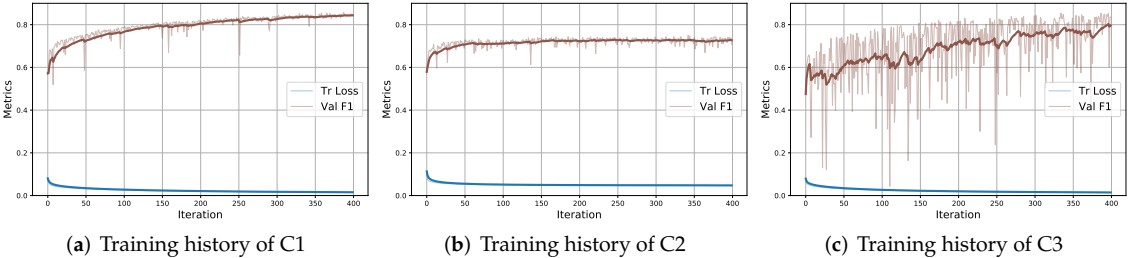

(**a**) Training history of C1    (**b**) Training history of C2    (**c**) Training history of C3

**Figure 5.** Evolution of training loss and F1 score on validation across the 400 training epochs on dataset $\mathcal{D}_2$. Bold lines indicate a running average with window of size 30.

Figure 5 depicts the evolution of the training loss and validation F1 score during training for the 3 configurations C1, C2, and C3. Please note that we do not plot the training accuracy, which is always above 98% since the first epochs, and we did not compute the F1 score on the training set. The latter requires the computation of predictions of the whole training set at each epoch and, given the size of the dataset $\mathcal{D}_2$, it would significantly prolong the training time that is already in the order of days.

From the plots in Figure 5, we notice that the training procedure is more stable when the OFCN is equipped with the SE module and achieves a higher F1 score when using BN. This might suggest that SE impacts model performance more than BN. Most importantly, none of the models is overfitting on the training set and the F1 score is still improving after 400 epochs. This means that that the training has not converged yet and better performance could be achieved by training the OFCN model for more epochs.

As a final experiment, we trained OFCN(32) configured with C1 for 500 epochs on down-scaled images (phase 1) and then for 3000 epochs on the full resolution images (phase 2). Training this model took almost two months but we obtained a significant improvement, reaching an exceptional 0.892 F1 score on the validation set. Figure 6 reports the training statistics for the second training phase, showing that after 3000 epochs the F1 score has finally converged.

We comment that a neural network with a smaller capacity could be trained in less time, but this would be traded with a lower segmentation performance. Indeed, we notice that even after so many training epochs (3000) the proposed deep-learning model is still not overfitting the training data. This suggests that the OFCN is not oversized with respect to the complexity of the dataset and the task at hand. Second and most importantly, we notice that once trained, the OFCN can segment a new large SAR scene within the order of seconds (A new $6400 \times 6400$ pixels image can be segmented in less than 1 minute on an Nvidia RTX 2080 when applying test time augmentation).

Figure 7 reports some examples of segmentation masks predicted by the OFCN on the validation set of $\mathcal{D}_2$. From left to right, we depict the VV input channel, the ground-truth mask made by human operator, and the OFCN prediction thresholded at 0.5 (values $\leq 0.5 \rightarrow 0$, values $\geq 0.5 \rightarrow 1$). To facilitate the interpretation of the results, we generated a bounding box around both the oil spills in the mask generated by the operator and the mask predicted by OFCN. If two bounding boxes overlap, they are *True Positives* (TP) and are colored in green. Bounding boxes appearing only in the predicted mask are *False Positives* (FP) and are depicted in blue. Finally, *False Negatives* (FN) are the red boxes appearing only in the human-made label but not in the prediction.

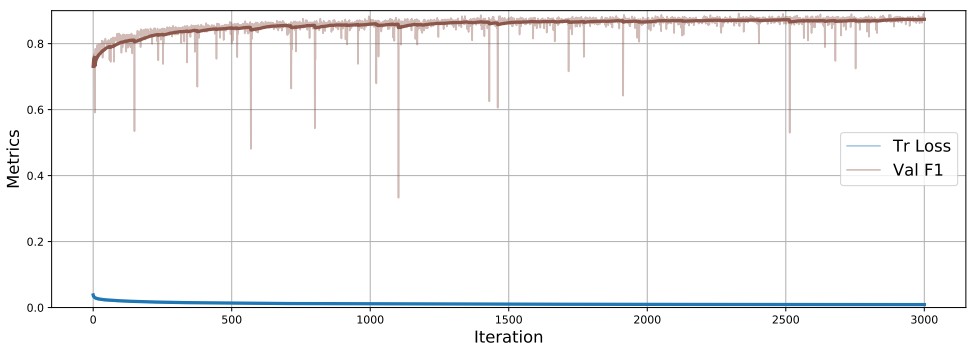

**Figure 6.** Training history of the model configured with C1 with two-stage training (C1-2ST-Long) on dataset $\mathcal{D}_2$. The plot depicts the evolution of the training loss and F1 score on the validation set over the 3000 epochs in the second stage. Bold lines indicate a running average with window of size 30.

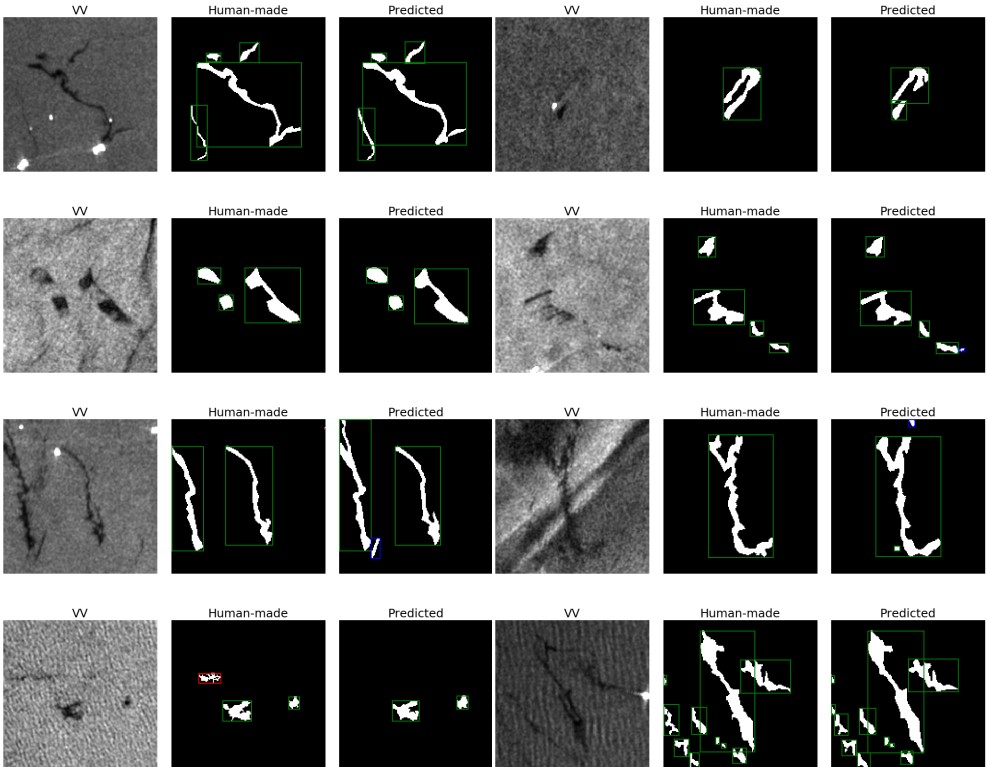

**Figure 7.** Examples of segmentation masks predicted by the OFCN on the validation set of $\mathcal{D}_2$. From left to right: the VV input channel, the mask made by the human operator, the OFCN output thresholded at 0.5 (values $\leq 0.5 \rightarrow 0$, values $\geq 0.5 \rightarrow 1$). Green bounding boxes are TP (the oil spill appears both in the human-made and the predicted mask), blue boxes are FP (the OFCN detects an oil spill that is not present in the human-made mask), and red boxes are FN (the oil spill is in the human-made mask but is not detected by the OFCN).

FPs are more common as they can arise from small details that might be overlooked by the human operator and often appear on the edges of the oil spill outline. On the other hand, FNs are very rare meaning that our model misses very few of the human-detected oil spills. Having a low number of FNs is particularly important because FP can always be discarded during a post-analysis, whereas a missed detection cannot be recovered.

From now on, by OFCN we will refer to the OFCN(32) model trained with configuration C1-2ST-Long (see Table 6).

### 5.1.5. Segmentation Performance and Incidence Angle

Here, we investigate how much the incidence angle of the satellite affects oil spills detection. All the patches in our dataset are characterized by an incidence angle between 30 and 45 degrees, which is the range where the scanSAR mode of Sentinel-1 operates [6]. Our initial hypothesis was that oil spills are detected more easily at medium inclinations since high and low incidence angles yield low oil-sea contrast [12,13]. However, the results disproved our hypothesis. In Figure 8, the red line shows the mean F1 score obtained in detecting oil spills at a given incidence angle; the red area shows the standard deviation; the blue bars indicate the number of oil spills for each incidence angle. To assess if there are statistical differences between the F1 scores obtained at different incidence angles, we perform the Kruskal-Wallis H-test, which is a non-parametric version of ANOVA that makes only a few assumptions about the characteristics of the population from which the data originate [37]. With the Kruskal-Wallis, we test the null hypothesis that the population median of all of the groups are equal and we obtain H-statistic = 54.14 and $p$-value $< 10^{-4}$. Since the $p$-value is much lower than 0.05 the null hypothesis cannot be rejected, meaning that there is not a statistically significant difference between the F1 scores obtained at different incidence angles.

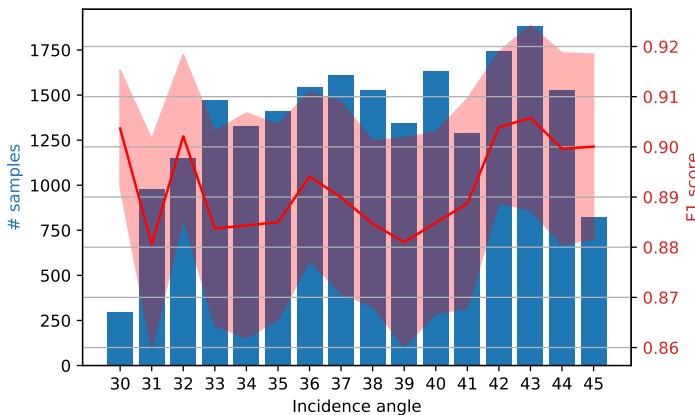

**Figure 8.** Variation of the F1 score on the validation set according to the incidence angle of the satellite.

However, we recall that the incidence angles available are in the range 30 to 45 degrees, where oil spill detection is usually preferred and the Bragg scattering dominates [12,38]. Specular reflection from both oil slicks and clean sea occurs outside this range, resulting in lower oil-sea contrast [12,39]. It is also interesting to notice that there are fewer oil samples at the near and far incidence angles (30 and 45 degrees), where oil slick detection is more challenging.

### 5.1.6. Visualization of the Learned Filters

To provide an interpretation of what the OFCN is learning, we synthetically generated images with patterns that maximally activate the convolutional filters in OFCN. To generate such images, we rely on the Activation Maximization technique [40]: we start from an input image containing random noise and we gradually update it by ascending the gradient $\frac{\delta ActivationMaximizationLoss}{\delta input}$.

Figure 9 depicts the patterns that maximally activate the first 64 of the 512 filters in the first convolutional layer of the "Enc Block 512" (the one at the bottom of OFCN in Figure 3). Interestingly, it is possible to notice several wave patterns, meaning that the OFCN is looking for waves when trying to detect the oil spills. This is reasonable since ocean surface waves are present in moderate wind conditions, which are the most favorable ones for detecting oil spills [41,42]. In fact, it is very difficult to detect oil spills if the sea surface is too flat due to low wind or is too irregular due to high wind. For low winds, the backscatter response is similar between the calm ocean surface and oil slicks [43]. However,

for strong wind, the oil can break and/or sink due to upper surface layer turbulence. The wind speed range for optimal oil-sea contrast is suggested to be 2–3 m/s to 10–14 m/s [41,42].

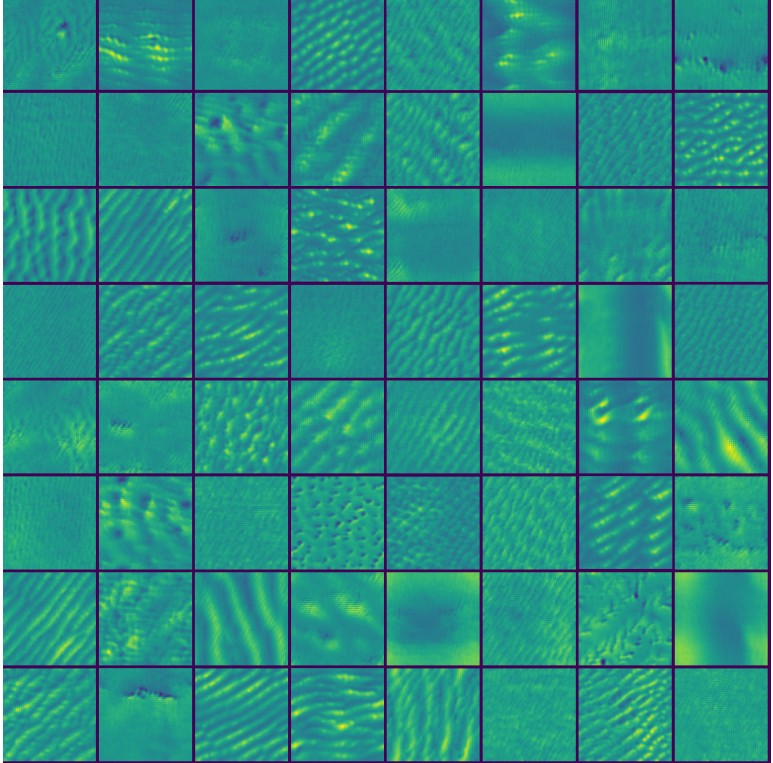

**Figure 9.** Filters visualization. We synthetically generated the input images that maximally activate 64 of the 512 filters in the first convolutional layer of the "Enc Block 512" at the bottom of the OFCN.

5.1.7. Results on the Test Set $\mathcal{D}_{\text{test}}$

Table 7 reports the performance obtained on the three test products of $\mathcal{D}_{\text{test}}$, whose details are in Table 3. Compared to the validation set, the proportion of oil pixels is much lower in the three SAR products. Also, given the large size of the images that extend up to 860 km, there is a higher chance of detecting FPs. For these reasons, the F1 scores Table 7 are lower compared to the score obtained on the validation set (0.892). We note, however, that a large portion of FPs is due to small bounding boxes not marked by the human operators.

**Table 7.** Performance obtained on the 3 SAR products in $\mathcal{D}_{\text{test}}$ used as test set.

| ID | F1 | IoU | TP | FP | FN |
|----|------|------|----|----|----|
| T1 | 0.73 | 0.81 | 2  | 7  | 0  |
| T2 | 0.44 | 0.36 | 11 | 59 | 0  |
| T3 | 0.83 | 0.52 | 31 | 14 | 5  |

As discussed at the beginning of this section, OFCN returns a soft output in [0,1] which must be rounded to obtain a binary segmentation mask. By varying the rounding threshold $\tau$ it is possible to vary quite significantly the number of TP, FP, and FN and the value of the two performance metrics, F1 score and IoU. In particular, with a lower $\tau$ more FP appear, while a higher $\tau$ implies more FN. From Figure 10 we observe that in the three test products by using higher $\tau$ the number of FP decreases significantly and the FN only increases in the third image (Figure 10c). This suggests that a high $\tau$ value improves the detection precision. On the other hand, the IoU and F1 score become much worse for high $\tau$ values, indicating that better contours can be found when using a lower threshold.

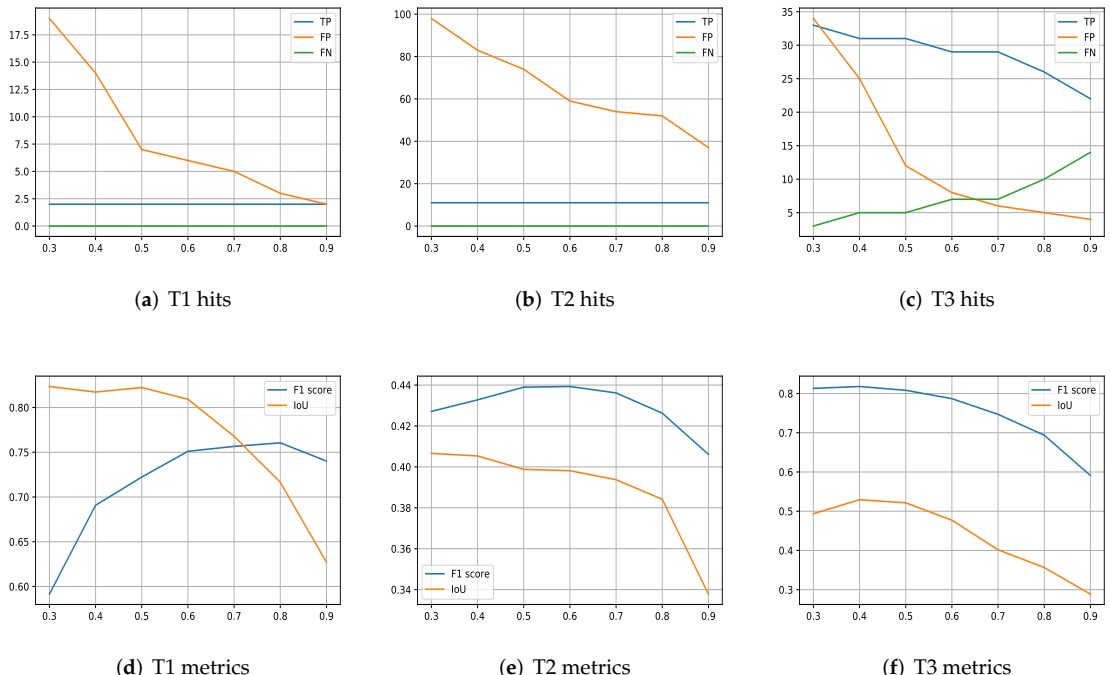

(**a**) T1 hits  (**b**) T2 hits  (**c**) T3 hits

(**d**) T1 metrics  (**e**) T2 metrics  (**f**) T3 metrics

**Figure 10.** The x-axis always denotes the value of the rounding threshold $\tau$. (**a**–**c**) number of True Positive (TP), False Positive (FP), and False Negative (FN) detection obtained by using a different threshold $\tau$ on the soft output of the OFCN. (**d**–**f**) values of F1 score and IoU for different $\tau$.

We will exploit the behaviors observed for different $\tau$ when implementing our visualization tool, presented in Section 6.

Figure 11 depicts examples of segmentation results on each one of the 3 SAR products in $\mathcal{D}_{\text{test}}$. Note the presence of a large FP in the second product (Figure 11f).

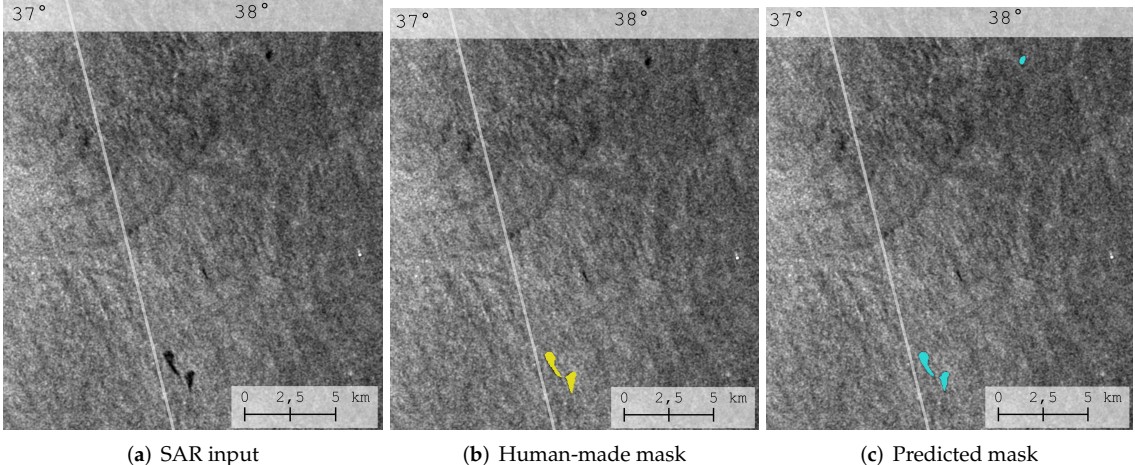

(**a**) SAR input  (**b**) Human-made mask  (**c**) Predicted mask

**Figure 11.** *Cont.*

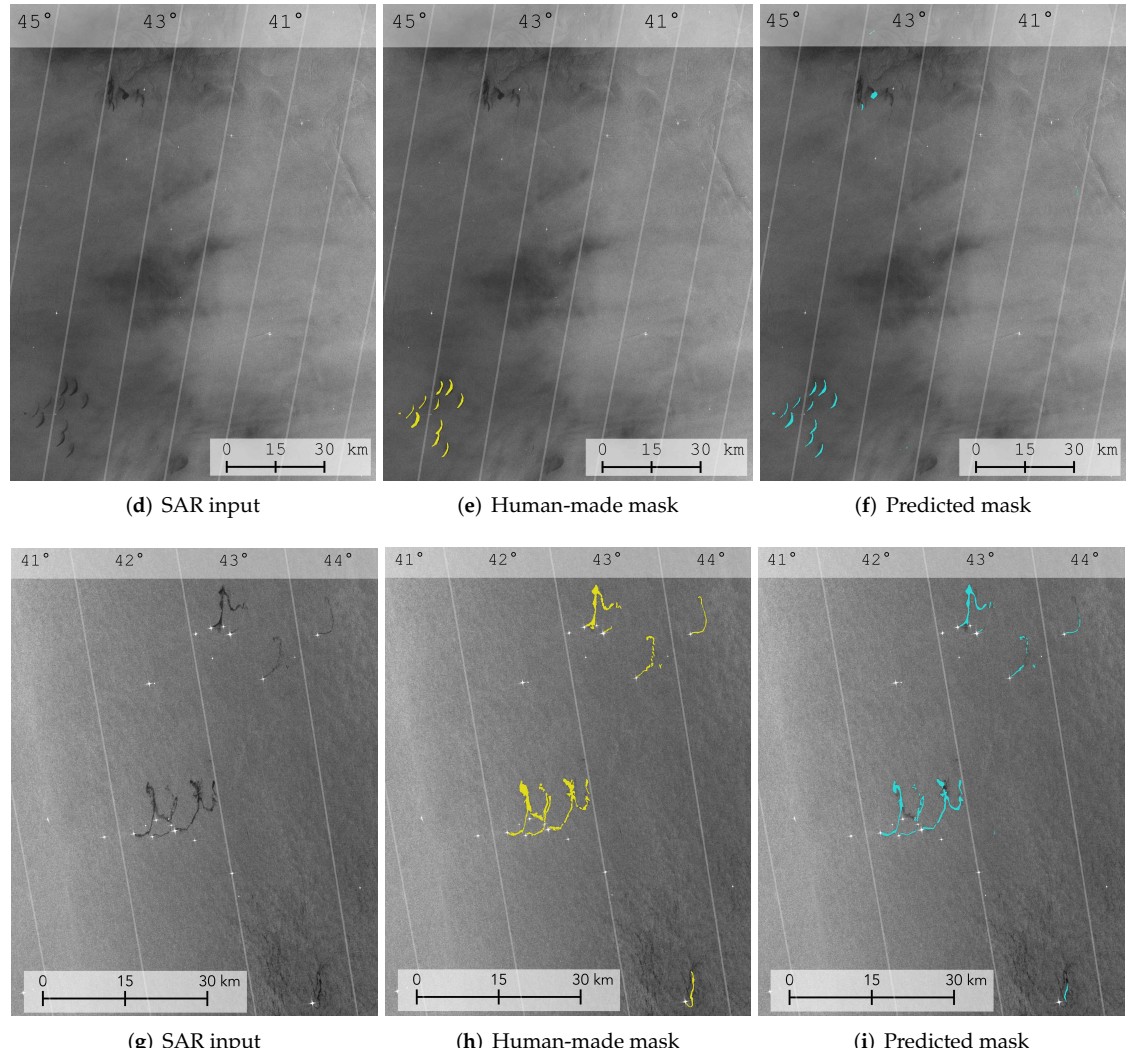

**Figure 11.** Results on the 3 Sentinel-1 products used for testing. The first column contains the original SAR images (VV-intensity); the second column contains the segmentation masks of oil spills that are manually drawn by human operators (yellow masks); the third column contains the segmentation masks of oil spills that are predicted by the OFCN (blue masks). Only small sections of the whole SAR products are shown in the figures. The number on the top of the images represent the incident angle.

*5.2. Oil Spill Categorization: Experimental Setting and Results*

To train the categorization network we first generate the predicted masks for both the training and validation set of $\mathcal{D}_1$ using the trained OFCN(32). We decided to use the soft predictions, i.e., we do not threshold the output of the OFCN with $\tau$ since it introduces an unnecessary bias and can conceal potential information of interest. As previously discussed, the soft OFCN output values are in [0, 1] and they can be interpreted as the *amount of certainty* that a pixel belongs to the oil class. Such a classification probability is lower in areas close to the edges of the oil spill and where the oil starts to dissolve.

We trained 12 different instances of the categorization models described in Section 4, one for each category. Each model is trained for 1000 epochs using Adam optimizer with initial learning rate $10^{-4}$, batch size 32, $L_2$ norm regularization weight $10^{-6}$, and dropout 0.1. Image augmentation is used with the following parameters: max rotation $90°$, max-width shift 0.1 of total width, max height shift 0.1 of total height, max shearing 0.3, max zoom 0.2, probability of horizontal and vertical flips 0.5, pad mode "mirror".

The accuracies obtained for each category are reported in Table 8. Since the values in most categories are unbalanced, we also report the F1 score as a performance measure.

**Table 8.** Classification accuracy and F1 score for each one of the 12 categories on the validation set.

| Category | Accuracy | F1 |
|---|---|---|
| Patch shape | 80.0% | 0.80 |
| Linear shape | 76.8% | 0.77 |
| Angular shape | 93.2% | 0.91 |
| Weathered texture | 70.4% | 0.64 |
| Tailed texture | 78.4% | 0.73 |
| Droplets texture | 98.8% | 0.98 |
| Winding texture | 94.4% | 0.92 |
| Feathered texture | 97.2% | 0.96 |
| Shape outline | 93.8% | 0.91 |
| Texture | 55.6% | 0.49 |
| Contrast | 61.6% | 0.59 |
| Edge | 61.6% | 0.58 |

Compared to traditional image processing tools, CNNs usually achieve very high performance on recognizing textures and they exploit this capability to achieve high accuracy in downstream classification tasks. However, in our case, the performance obtained for some texture categories are particularly low. We argue that one of the reasons is the presence of noise in the labeling process since it is difficult to precisely determine texture and contrast features from a SAR image in a consistent manner. We also observe a low accuracy and F1 score in some other categories, such as "Contrast" and "Edge". Compared to the masks in the segmentation task, the classification labels are less reliable since the labeling procedure is more subjective and there is room for human errors. Most importantly, the trained operators use complementary information to define the categories, such as sea state, wind, and historical seep sites. The operators also account for nearby potential polluters (ships/platforms), by combining the automatic identification system (AIS) and sea maps. Since all this information are not contained in the SAR products, a classification model based only on the image content can struggle in determining the right category.

## 6. Large-Scale Visualization

We developed a pipeline to automatically acquire all the SAR products available in a given area and within a specified time frame and then process them with our deep-learning framework. Our pipeline performs the following steps:

1. as input, we only specify the coordinates of an area and the time interval of interest;
2. all the SAR products within the time frame that overlaps at least 20% with the specified area are fetched from the Alaskan SAR Facility (ASF) repository;
3. since the SAR images come from Sentinel-1 (GDRH, 10 m resolution), they are first smoothed and then downsampled by a factor of 4 to match the mode of our training data;
4. all the SAR products are processed with the OFCN described in Section 3; the procedure consists of two steps, *filtering* and *coloring*, discussed below;
5. for each oil spill, we generate a vector of features that includes the size of the slick and the distance from the closest oil spill detected;
6. very small slicks are discarded, i.e., slicks whose surface is lower than 0.25 km$^2$ and are farther than 1.5 km from any other detected oil spill.

Recalling the discussion related to Figure 10, it is possible to obtain high precision in the detection by thresholding the OFCN output with a high $\tau$. On the other hand, with a smaller $\tau$ the oil spill contours are more accurate. For this reason, we first perform a *filtering* step by applying $\tau = 0.8$ to keep only larger slicks and discard many FPs. In a second *coloring* step, we compute with $\tau = 0.5$ the

outline of the slicks that remain after filtering. The *filtering-coloring* procedure is very fast since the soft output of OFCN does not need to be recomputed.

The results obtained are encoded into geojson files they are visualized with NLive (http://nlive.norut.no), our geographic visualization tool. In NLive it is possible to select the individual oil slicks and visualize a small chunk of SAR image where the oil spill has been detected, plus additional information about shape and the distance from the closest neighbor. The oil spills detected in an area of approximately 100,000 km$^2$ in the South hemisphere between October 2014 and March 2020 are shown in Figure 12. In the example, a total of 501 SAR products were retrieved, and in 136 of them at least one oil spill has been detected; a total of 665 oil spills were found.

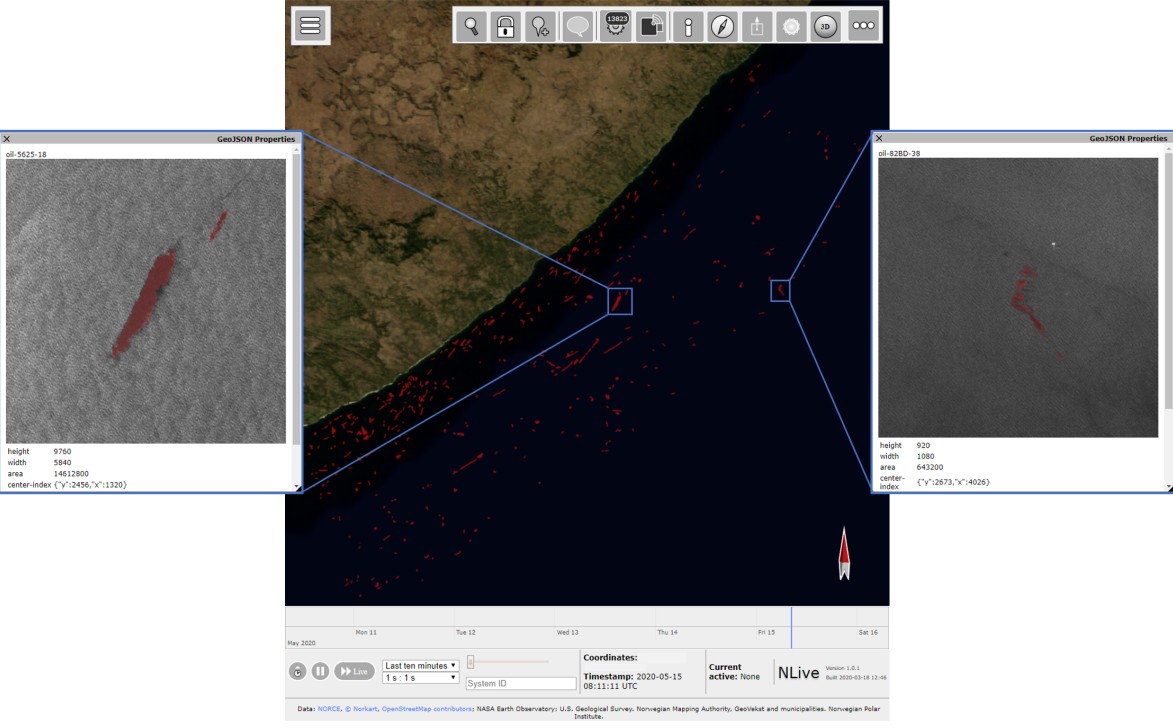

**Figure 12.** Visualization in NLive of oil spills detected in a large area (approximately $500 \times 200$ km$^2$) in the South hemisphere between 2014 and 2020.

## 7. Conclusions

In this paper, we proposed a deep-learning framework to perform detection and categorization of oil spills on a large-scale. We formulated oil spill detection as an image segmentation task, where each pixel in an input SAR image is assigned to the class "oil" or "non-oil". We designed a fully convolutional neural network for semantic segmentation, which we trained on pairs consisting of a small patch of a large SAR product and an associated binary mask, drawn by a human operator, that defines the class for each pixel. Through an extensive experimental evaluation, we demonstrated the capability of the proposed architecture in achieving high detection performance, obtaining results comparable to human operators. The proposed framework is trained on detecting mineral oil slicks mostly originating from human releases. Look-alikes are present in our dataset, but they are labeled as "non-oil" rather than being assigned with a specific class. Since a high F1-score was obtained, look-alikes are rarely detected as mineral oil slicks, meaning that the proposed deep-learning approach manages to distinguish well between them.

Once the oil spill is detected, we used a second neural network to predict information about its shape, texture, and contrast according to 12 different categories. Differently from the detection task, this categorization is not done at a pixel level but is relative to the whole patch. Our is the first exploratory work in categorizing oil spills in SAR images. The categorization results are useful

to end-users and analysts to derive information about the source, stage of weathering and internal variations within the oil slicks that could be related to oil concentration or thickness, and also to help distinguishing oil slicks from look-alikes.

Despite neural networks are particularly capable of detecting textures, we obtained a low classification accuracy for some of the categories. We believe the main reasons are the use of exogenous data to determine the category, but also noise and inconsistency in the human-made labels. Indeed, it is extremely difficult to precisely determine texture and contrast features from a SAR image in a consistent manner. Remarkably, our findings on the automatic categorization performance provided valuable insights for improving the design of future oil spill services by operators such as KSAT.

Finally, we presented a production pipeline to detect and visualize the presence of oil spills worldwide at given times in history. Our pipeline fetches SAR products from the ASF repository of Sentinel-1 images and performs automatic detection and categorization. The results are visualized in an interactive geographical map, where each oil spill can be individually selected to be further analyzed. To the best of our knowledge, this is the first tool based on deep learning that allows analyzing oil activity on such a large scale.

**Author Contributions:** Contributions F.M.B. was responsible for the machine learning methodology: he designed the deep-learning architectures and performed the experiments. M.M.E. was responsible for the remote sensing part: she provided the background on oil spills detection, she guided the experimental evaluation and interpretation of the results. N.B. developed the production pipeline that integrates the proposed deep-learning framework and the NLive visualization tool. F.M.B. and M.M.E. wrote the paper. All authors have read and agreed to the published version of the manuscript.

**Funding:** The research done by M. Espeseth was funded by CIRFA through the Research Council of Norway (Norges forskningsråd), research grant number 237906. The research done by F. M. Bianchi and N. Borch was funded by KSAT (Kongsberg Satellite Services), which is leading the Gonzales project (research grant number 282082) within the PETROMAKS 2 program of the Research Council of Norway (Norges forskningsråd). The publication's cost of this research was funded by KSAT.

**Acknowledgments:** We acknowledge the work of the following researchers at NORCE: Ingar Artnzen processed and prepared the SAR dataset for training the deep learning models; Per Egil Kummervold contributed to the discussion about the design of the deep-learning methods; Daniel Stødle is the main developer of NLive, the tool used in this study for large-scale visualization of oil spills detected worldwide.

**Conflicts of Interest:** The authors declare no conflict of interest. One of the funders, KSAT, participated to the discussions about the design of this study and provided us with the extensive dataset of manually annotated oil spills in SAR images, which has been used in this study to train and evaluate the deep learning models. KSAT approved the discussion and the analysis of the results and agreed to the decision of publishing the results obtained in this study. Finally, we report that the source of the Sentinel-1 data is Copernicus Sentinel data, retrieved from ASF DAAC and processed by ESA.

## Appendix A. Further Details of the Segmentation Model

### Appendix A.1. Bilinear Upsampling Layer

A standard 2D upsampling procedure enlarges the image simply by inserting new rows and columns between the existing ones and fills them by replicating the content of neighboring pixel values. Instead, to obtain a more accurate generation of the output map, we perform bilinear upsampling in the decoder layers of the OFCN. Bilinear upsampling computes the new pixel values by performing a linear interpolation between the adjacent original pixels. It has been shown that bilinear upsampling yields a more accurate reconstruction and the architecture equipped with it obtains a better segmentation accuracy [44].

### Appendix A.2. Batch Normalization Layer

To speed up the convergence of the training and provide a regularization to the network that improves its generalization capabilities, we applied Batch Normalization (BN) [29] before each non-linear activation in the OFCN, both in the encoder and decoder. BN normalizes channel-wise the mean and the range of the values of the activations in the previous layer, making the network output

almost invariant to the scaling of the activations of the previous layers. BN has the effect of reducing the variance in the distribution of layer activations during training, preventing the network weights to diverge and the activation to saturate. Empirically, it was shown that BN stabilizes and accelerates the training while reducing the need to tune a variety of other hyperparameters to achieve higher performance [45].

*Appendix A.3. Squeeze-and-Excitation Layer*

We equipped the encoder of the OFCN with SE modules, which improve channel interdependencies at almost no additional computational cost [30]. SE gets a global understanding of each channel by squeezing each feature map to a single numerical value.

Figure A1 illustrates the mechanism of the Squeeze-and-Excitation block. Let $\mathbf{X} \in \mathbb{R}^{H \times W \times C}$ be a feature map generated, for example, by a convolutional layer, where $H$ and $W$ are the height and width of the features maps and $C$ the number of channels (i.e., the filters of the convolutional layer). A squeeze operation aggregates the feature map $\mathbf{X}$ across the spatial dimensions $H$ and $W$. The resulting embedding is a vector in $\mathbb{R}^C$ that captures the global distribution of channel-wise feature responses. An excitation operation uses the embedding vector to implement a self-gating mechanism that rescales the weights of the original feature map channel-wise. The resulting feature map $\tilde{\mathbf{X}}$ is used as input for the next neural network layer.

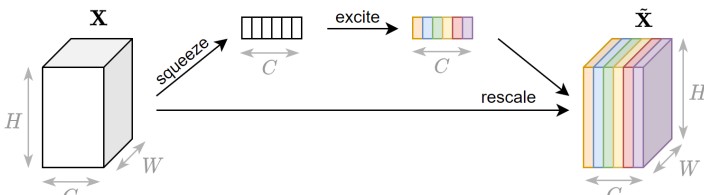

**Figure A1.** Overview of the Squeeze-and-Excitation block used in the OFCN encoder. The SE blocks are inserted after each ReLU activation.

*Appendix A.4. Receptive Field of the OFCN Architecture*

Figure A2 depicts the exponential growth of the receptive field across the layers of the encoder in the proposed OFCN architecture to perform segmentation. The diagram considers only the convolutional and pooling layers because are the only ones responsible for changing the size of the receptive field.

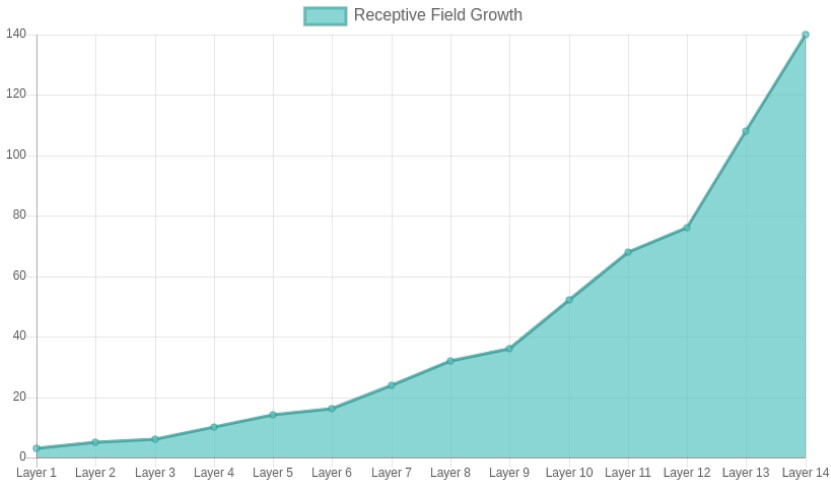

**Figure A2.** Growth of the receptive field in the layers of the encoder.

## Appendix B. Additional Experimental Details

*Appendix B.1. Training Stats of Deep-Learning Architectures*

Figure A3 depicts the evolution of the loss on the training set and the F1 score on the validation set during training, for the three deep-learning models (U-net, DeeplabV3+, and the proposed OFCN) compared in Section 5.1. The models are trained on the dataset $\mathcal{D}_1$. The plots show that none of the models is overfitting on the training set after 400 epochs.

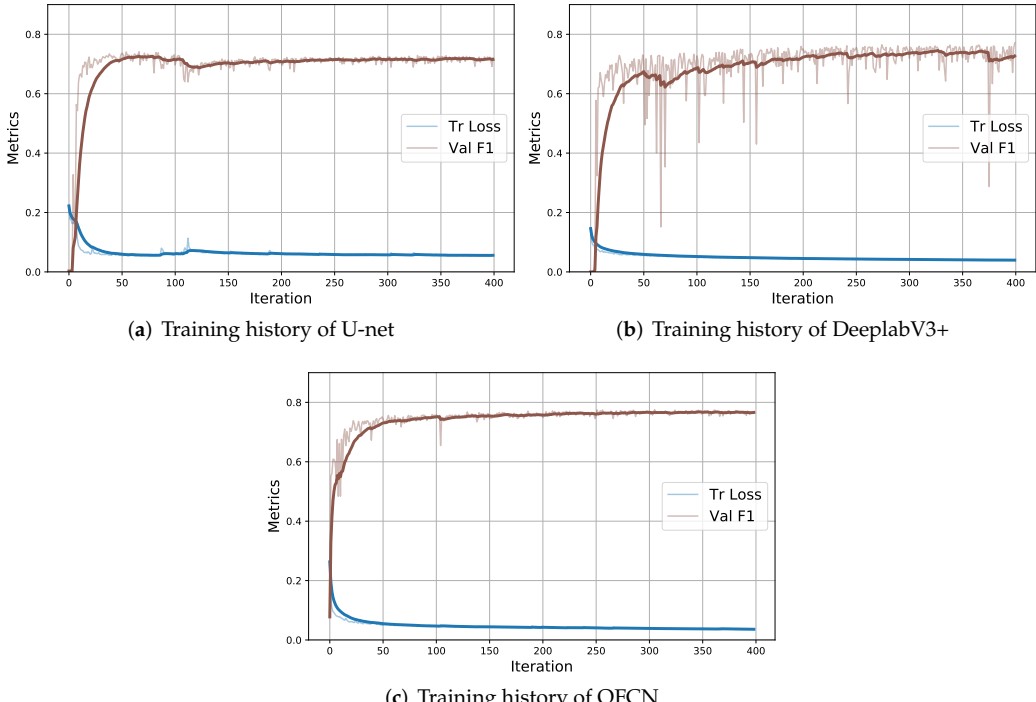

(**a**) Training history of U-net　　　　　　　　　　(**b**) Training history of DeeplabV3+

(**c**) Training history of OFCN

**Figure A3.** Evolution of training loss and F1 score on validation across the 400 training epochs on dataset $\mathcal{D}_1$. Bold line indicates a running average with window of size 30.

## Appendix C. Unsuccessful Approaches

In the following, we mention other strategies we experimented with but did not provide satisfactory results.

*Appendix C.1. Loss Functions for Class Imbalance*

Besides the binary cross-entropy with class balancing, we tried three additional loss functions that are specifically designed to handle classes with an uneven number of samples.

- *Focal loss.* Addresses class imbalance by reshaping the standard cross-entropy loss such that it down-weights the loss assigned to well-classified examples [46]. The Focal Loss focuses on training on a sparse set of hard examples and prevents the vast number of easy negatives from overwhelming the detector during training. The Focal Loss is defined as

$$FOC(p_t) = -\alpha(1 - p_t)^\gamma \log(p_t). \tag{A1}$$

  We used the default parameters $\alpha = 0.25$ and $\gamma = 2$ proposed in the original paper [46].
- *Jaccard Loss* handles class imbalance by computing the similarity between the predicted region and the ground-truth region for an object present in the image. In particular, the loss penalizes

a naive algorithm that predicts every pixel of an image as the background, as the intersection between the predicted and ground-truth regions would be zero [47]. The Jaccard loss is defined as

$$JAC(X,Y) \ = \ \frac{|X \cap Y|}{|X| + |Y| - |X \cap Y|} \ = \ \frac{\sum |X \odot Y|}{\sum |X| + \sum |Y| - \sum |X \odot Y|}, \tag{A2}$$

where $\odot$ indicates the Hadamard product.

- *Lovász-softmax loss* is an extension of the Jaccard Loss, which generates convex surrogates to submodular loss functions, including the Lovász hinge. We refer to the original paper for the formal definition [48]. The official TensorFlow implementation (https://github.com/bermanmaxim/LovaszSoftmax) of the Lovász-softmax loss has been used to perform the experiments.

For each loss function, we repeated the same hyperparameters search described in Section 5.1 and in Table A1 we report the best configuration found and the associated F1 score.

**Table A1.** Best configurations and F1 scores for loss functions different from binary cross-entropy. *Acronyms:* BN (Batch Normalization), SE (Squeeze-and-Excitation), $L_2$ reg. (strength of the $L_2$ regularization on the network parameters), LR (Learning Rate), FOC (Focal loss), JAC (Jaccard loss), LOV (Lovász-softmax loss).

| Loss | BN | SE | $L_2$ reg. | Dropout | LR | F1 ($\mathcal{D}_1$) |
|------|------|------|------------|---------|------|--------|
| JAC | True | True | $1 \times 10^{-6}$ | 0.1 | $1 \times 10^{-3}$ | 0.667 |
| FOC | True | True | $1 \times 10^{-3}$ | 0.0 | $1 \times 10^{-2}$ | 0.664 |
| LOV | True | True | $1 \times 10^{-5}$ | 0.0 | $1 \times 10^{-4}$ | 0.597 |

It is immediately possible to notice that the results are significantly lower than those reported in Table 4 and obtained by using binary cross-entropy with class-specific weights. In particular, when optimized with the Lovász-softmax loss, our model achieves an F1 score that is 17% lower than the F1 score obtained with the weighted binary cross-entropy.

*Appendix C.2. Conditional Random Field*

The prediction of the segmentation mask can be modified by using CRF [49] as a subsequent post-processing step. CRF produces a result that is given by the combination of the pixel-wise neural network prediction, the pixel value in the input image (SAR value in this case) and the pixel position. More formally, the network prediction $\psi(x_i)$ for pixel $i$ is combined with the following pairwise potential

$$w^{(1)} \exp \left( -\frac{|p_i - p_j|^2}{\theta_\alpha^2} - \frac{|I_i - I_j|^2}{\theta_\beta^2} \right) + w^{(2)} \exp \left( -\frac{|p_i - p_j|^2}{\theta_\gamma^2} \right) \tag{A3}$$

where $j$ are the indices of the other pixels in the patch, $p_i$ indicates pixel position and $I_i$ the SAR value of pixel $i$. The parameters configuration used for the training is $w^{(1)} = 5, w^{(2)} = 0.1, \theta_\alpha^2 = 2, \theta_\beta^2 = 2$, and $\theta_\gamma^2 = 1$.

We found CRF to be computationally intensive, very sensitive to the values of the hyperparameters, and, most importantly did not bring a significant improvement in the segmentation performance. Similar results were found also in other related work [50].

*Appendix C.3. Multi-Head Classification Network*

In a first attempt, to perform the categorization task we designed an architecture that shares the first 5 convolutional blocks and has 12 different classification heads, each one specialized in predicting one of the 12 categories. Such a network can predict at the same time all the categories given the input

VV and mask. However, we achieved a better accuracy by training 12 different networks independently with a single head (one for each category) such as the one depicted in Figure 4.

*Appendix C.4. Gradient Descent Optimizers*

As an alternative to Adam, the Nadam optimizer [51] minimizes the loss faster in the first epochs, but in the end, settled worse to minima than Adam.

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
