# Peer review of "Large-Scale Detection and Categorization of Oil Spills from SAR Images with Deep Learning"

_remotesensing, doi:10.3390/rs12142260_

Round 1
Reviewer 1 Report
The paper addressees a very important task to apply computer vision and deep learning. The numbers presented to show the importance of the oil spill detection on the sea are a good argument to remark the benefits of the study. The visualization tool is also a useful outcome of this research. I consider the approach in general a good way to conduct the task, while some elements have to still be clarified for the readers. The features of the datasets as well was the training parameters require some additional information. I mention some suggestions in the following points:
1. In line 16 says “both”, while there are three options given.
2. In line 86 there is a second reference to section 3 instead of section 4.
3. In line 101 there is a good description about the images source. It doesn’t mention any specific preference for a geographic region. Are these taken from different sites around the world?
4. Although the clipping of the data to include the 98% percentile (line 128) is a good way to reduce the effect of the outliers, does not it reduce as well the range for the non-oil part? (Using 180 or 200 as upper threshold might keep some features to separate non-oil pixels).
5. Is there a reason to select the patch size (line 142)? Most of baseline frameworks can work with higher resolutions. Hence, larger patches might find more context around the oil spills.
6. For the table 1, which is the reference for the units used? Previously the backscatter response is given in the [0,150] interval. What do these new values in the table represent?
7. In section 2.2, why is not mentioned a test set for D2?
8. For the table 3, does the pixel size column refer to the source image size in terms of pixels or the meters covered on the surface that each pixel represent?
9. The section 3.2.3 proposes an option to refine the information. This is similar to the ResNet backbone models, which retrieve finer details. Was this option taken into account? A second stage training could be avoided.
10. In line 301 the bounding boxes are mentioned. How are these computed by the network? So far the network was described to compute identification (binary) and classification (softmax). The part for the bounding boxes should be described with more detail. Also its inclusion in the hand-made ground truth.
11. In the table 4 the use of SE seems to have a better effect than BN, was that found in many other configurations as well? This could be an interesting result to report.
12. In the line 335 is reported that the U-net did not learn anything in many runs. Was any algorithm used to initialize weights in the networks? These algorithms usually prevent the vanishing or exploding trends in the weight values.
13. The best case listed in the table 6 took almost two months to achieve a higher accuracy. This time is not feasible to run many experiments. Maybe changes in the network can lead to such a good performance while reducing the training times.
14. In the line 353 it is mentioned that the network achieved 98% accuracy, but in the line 228 was said that such performance can be obtained by a naive classifier. Is the network performing worse than a naive classifier in the first epochs?
15. The conclusions obtained from the image 9 are a good outcome to present. It also implies the advantages of using deep learning.
16. Presenting results where patches from the dataset do not perform optimally would be useful. This can lead to better suggestions to tackle challenging patches by observing the features they share.
17. At the beginning, it is mentioned that many images can lead to confusions for oil spills (like the biogenic slicks). Are images of these nature also part of the dataset?
Author Response
Dear reviewer 1,
please find attached the pdf that replies to all your comments.
Best regards,
Filippo Bianchi

Reviewer 2 Report
Dear authors I read your paper with great interest, and found it interesting to a point. I feel this work is interesting, but the authors stumble into a key aspect of oil in the sea when discussing their method. Some of the oil present on continental margins around the world is naturally seeping through cold vents, slope seeps, and faults, and one of the ways of finding new hydrocarbons is by mapping the natural seep of oil into the sea. I am not saying that the method presented in this work is wrong, but it may provide a over-interpretative tool to identify pollutants in the sea, when they are the expression of natural processes. I stress that one of the main ways to understand if an active hydrocarbon system occurs offshore is by mapping these natural seeps - as it happens onshore, to be more precise. Therefore, human input is necessary at the end of your method, and my view is that the authors should try to present - in the discussion - ways to discern natural from man-made spills. I have annotated a .pdf with a computer stylo (a electronic pen that allows me to write on .pdfs) and the numbers below refer to parts of the text: Line 15 - References are needed at the end of the first statement in the introduction, Point 1 - Lines 17 to 18 - Having analysed the (public) data available for the North Sea under the NEREIDs project (Nereids.eu), I find this statement very overblown. From the 200+ platforms in the North Sea, on average, each releases 1.5 cubic meters of dilutant and chemical products onto the sea per year. These are normally the result of brief spills and are an average - this means that they are much less that 1.5 cubic meters/year as they include the larger spills in the calculations. These larger spills are usually ruptures of pipelines or minor leaks often caused by fishing trawlers. I feel that 1.5 cubic feet of a dilutant, per year, per platform, is a very small amount to consider as a major pollutant. Point 2 - Look alikes are very common, as oil and gas seepage from the sea floor is part of the natural carbon cycle of this planet. Thus, can you distinguish the look-alikes from human spillage using your algorithm? Or do you need a person (and a comprehensive database) to distinguish one from the other? Points 3 and 4 - Conclusions - This part is very important, and shows that the authors did not address the fact that look-alikes should be distinguished from illegal spills. They are natural in terms of their genesis. Could the authors use the Discussion to address this point, before mentioning it in the Conclusions? All in all, this paper needs only a moderate review, and a better discussion.

Author Response
Dear reviewer 2,
please, find attached the PDF with the answers to all your comments.
Best regards,
Filippo Bianchi
